# Strong PUF Enrollment with Machine Learning: A Methodical Approach

**Amir Ali-Pour** [1,*,†], **David Hely** [1,*,†] ![ID], **Vincent Beroulle** [1,*,†] ![ID] and **Giorgio Di Natale** [2,*]

1   Grenoble INP, LCIS, University Grenoble Alpes, 26000 Valence, France
2   CNRS, Grenoble INP, TIMA, University Grenoble Alpes, 38000 Grenoble, France
*   Correspondence: amir.ali-pour@lcis.grenoble-inp.fr (A.A.-P.); david.hely@lcis.grenoble-inp.fr (D.H.);
    vincent.beroulle@lcis.grenoble-inp.fr (V.B.); giorgio.di-natale@univ-grenoble-alpes.fr (G.D.N.)
†   These authors contributed equally to this work.

**Abstract:** Physically Unclonable Functions (PUFs) have become ubiquitous as part of the emerging cryptographic algorithms. Strong PUFs are also predominantly addressed as the suitable variant for lightweight device authentication and strong single-use key generation protocols. This variant of PUF can produce a very large number of device-specific unique identifiers (CRPs). Consequently, it is infeasible to store the entire CRP space of a strong PUF into a database. However, it is potential to use Machine Learning to provide an estimated model of strong PUF for enrollment. An estimated model of PUF is a compact solution for the designer's community, which can provide access to the full CRP space of the PUF with some probability of erroneous behavior. To use this solution for enrollment, it is crucial on one hand to ensure that PUF is safe against a model-building attack. On the other hand, it is important to ensure that the ML-based enrollment will be performed efficiently. In this work, we discuss these factors, and we present a formalized procedure of ML-based modeling of PUF for enrollment. We first define a secure sketch which allows modelability of PUF only for a trusted party. We then highlight important parameters which constitute the cost of enrollment. We show how an ML-based enrollment procedure should use these parameters to evaluate the enrollment cost prior to enrolling a large group of PUF-enabled devices. We introduce several parameters as well to control ML-based modeling in favor of PUF enrollment with minimum cost. Our proposed ML-based enrollment procedure can be considered a starting point to develop enrollment solutions for protocols which use an estimated model of PUF instead of a CRP database. In the end, we present a use-case of our ML-based enrollment method to enroll 100 instances of 2-XOR Arbiter PUFs and discuss the evaluative outcomes.

**Keywords:** Physically Unclonable Function (PUF); strong PUF; PUF enrollment; machine learning (ML); artificial neural network (ANN)

## 1. Introduction

Security in internet-based communication, especially in IoT systems and Cyberphysical systems, is facing new opportunities and challenges as concepts such as Physically Unclonable Function (PUF)-based cryptographic methods are emerging. PUF is considered as one of the emerging security primitives for resource–constraint ecosystems in the field of IoT [1,2]. PUF is characterized as a hardware bound function which utilizes the unit-specific micro-variations to generate device-specific data. The functionality of PUF is based on mapping a bit-vector challenge (the input) to a response (output) and generating a so-called Challenge–Response–Pair (CRP).

Two variants of PUF exist so far, the strong PUF and the weak PUF. Strong PUF is a macro variant which aims at generating an abundance of device-specific identifiers which are products of the PUF CRP [3,4]. The weak PUF, on the other hand, is able to generate few CRPs (often only one, for memory-based PUFs) [5]. The former variant is mostly discussed for authentication protocols, and the lateral variant is known for the primitive source of encryption key generation [6–9].

Strong PUF is a variant of PUF that contains a very large CRP space [1]. Commonly, a strong PUF structure comprises a large challenge bit-vector $C$ as the input (usually 64-bit or 128-bit), which leads to $2^{sizeof(C)}$ combinations of different challenges in total, and, for each challenge vector, there is also a response value as the output of the PUF, which commonly is a binary value $r \in \{1, 0\}$. Collecting such number of CRPs is infeasible, mainly due to the shortage of memory space. Even for one PUF, it takes petabytes of storage space to contain all the CRPs.

Due to the large CRP space, strong PUF are potential sources to generate single-use device identifiers or encryption keys. For such applications, it is commonly imagined to employ multiple strong PUFs on a silicon chip, where the response of each individual strong PUF is binary [1,10], while the collocation/concatenation of the PUFs' response values is a multi-bit output with high entropy. Such multi-PUF sketch is potential for generating encryption keys that can be single-use. Given also that the CRP space of the constituting PUFs is very large, the system employing such sketch for key generation practically never runs out of single-use encryption keys. Although this is a specific use-case for strong PUF, it shows clearly that the potential exists with harnessing the large CRP space.

To deal with the shortage of storage space to contain all the CRPs of a strong PUF, it is potential to generate an estimation of strong PUF using Machine Learning modeling [2]. There already exists a matured research field for strong PUF modeling, which contains a large number of different techniques that tackle the PUF modeling in different ways [11–16]. Although these methods are mainly proposed to notify the designers of existing attack methods which aim at model-building the PUF, the benefit of ML-based modeling of PUF, however, is not limited to that use-case. Alternatively, model-building of strong PUF can be utilized by designers for device enrollment. This use-case has already been discussed in several works such as the Slender PUF authentication protocol proposed in [7], a mutual lightweight authentication method proposed in [17,18], and an encryption protocol proposed in [19].

Modeling strong PUF, however, can in turn be seen as a heavy task, requiring a large amount of CRP and PUF data to yield an accurate model. Depending on the level of PUF complexity, modeling PUF in turn can lead to requiring over a million CRPs. While capturing a large number of CRPs for one device is not an issue, practicing it for a large group of devices can be seen as a cost issue. The lateral is the setting that designers can face if they do not consider a cost-aware enrollment process.

In this work, our goal is to elaborate on the procedure of enrollment based on Machine Learning in order to define a cost-aware process with adjustable control parameters that in turn lead towards optimal enrollment with minimum cost. Here, we first draw the sketch for a PUF-based ecosystem, which is fit for ML-based PUF computing. This ecosystem should comprise the fundamental requirements in terms of reliability and security for ML-based PUF computing. We then focus on ML-based PUF enrollment and identify what constitutes the cost parameters and how the enrollment procedure can control the training to yield cost-efficient ML solutions. This procedure comprises an initial evaluative process that identifies the optimal specifications to carry the enrollment task on a large number of PUFs. We demonstrate a use-case for our ML-based PUF enrollment procedure on large number of 2-XOR Arbiter PUFs using simulated data. We show that the evaluative process is necessary to accompany an ML-based PUF enrollment task to identify the optimal parameter values for the learning phase in order to yield accurate models cost-efficiently.

We clarify here that our work is not competitive with the previous works, which practice ML-based PUF modeling. Moreover, the novelty of this work does not lay with the ML-based solution we practice. Instead, our work attempts to provide a mutual base ground for ML-based enrollment solutions.

In Section 2, we elaborate on the general ML-based PUF computing sketch. In Section 3, we explain some of the essential secure CRP access sketches. In Section 4, we explain the ML based modeling of strong PUF and highlight the important parameters for enrollment. In Section 5, we elaborate on our proposed ML-based PUF enrollment procedure and, in Section 6, we explain the method for strong PUF enrollment using our proposed ML-

based procedure. Section 7 comprises the experimental work. In the beginning of the section, we elaborate on our experimental setup. Later in the section, we discuss the results of performing our ML-based enrollment procedure on a set of 100 2-XOR Arbiter PUFs. Section 8 presents the conclusions of our work and the future steps.

## 2. Machine Learning and PUF-Based Computing

The general sketch of ML-based PUF utilization can be seen in Figure 1. Here, the first priority is to build an accurate model of PUF during the enrollment (see Figure 1a). Since the ML-based model is an estimation of the CRP characteristic of PUF, there is a probability that the model miss-predicts the response to a given challenge. Thus, it is crucial to decrease the miss-prediction probability as much as possible during the training process. In the meantime, access to the CRP of the PUF is open during enrollment. Here, the assumption is that the enrollment is done during the manufacturing phase, and the open CRP access through physical channels is given to speedup the CRP read-out process.

In mission mode (see Figure 1b), the ML model is used to provide CRP for the trusted party who is communicating with the PUF-enabled device. In this communication, the Server and the PUF-enabled device exchange several CRPs. Here, it is crucial that the PUF and the communication channel are secure. Thus, protection methods such as masking the response should exist to prevent the CRP leak, and consequently decrease the chance of obtaining an accurate model of the PUF for an unauthorized party who is collecting the transmitting CRPs. In the following, we review several secure sketches for accessing CRP of strong PUF.

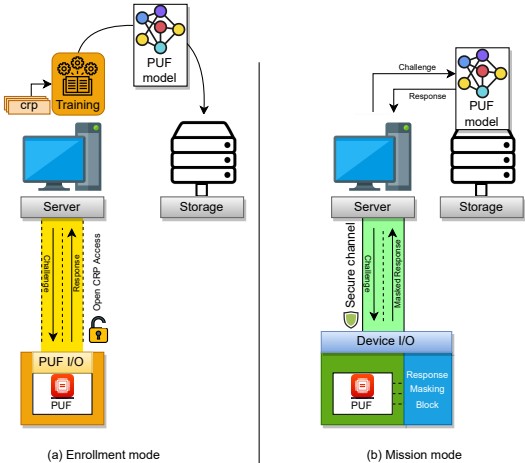

(a) Enrollment mode                    (b) Mission mode

**Figure 1.** Schematic of a communication between PUF and a verifier server, via an estimation model of PUF.

## 3. Secure CRP Access for Strong PUF

In communicating with a given PUF-enabled device, there needs to be a CRP exchange. There are two modes in which the CRP exchange happens. The primary mode is the enrollment where an enrolling server queries the PUF with random challenge values and obtains the responses. In this mode, the assumption is that the Server has direct access to the PUF via a physical I/O interface to read the CRPs (see Figure 2a). We assume that such access is given commonly during the test phase of the manufacturing process. Once the CRP read-out is complete, the physical access to the PUF should be permanently disabled (see Figure 2b). This means that there is no direct physical channel to communicate with the PUF circuit from outside of the device once the device is enrolled. An example this sketch has been discussed in Slender PUF protocol in [7]. Here, the authors propose using e-fuse mechanism to prevent having direct access to the PUF response after the enrollment phase. Accordingly, e-fuses are disabled once the PUF is enrolled, and thus the PUF responses are no longer accessible through the chip I/O.

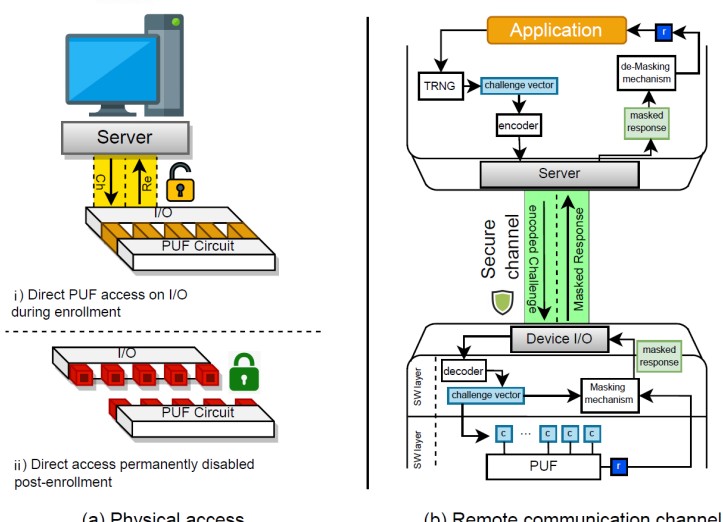

(a) Physical access      (b) Remote communication channel

**Figure 2.** Secure sketches for PUF-based Computing. Here, (**a**) demonstrates the permanently disabling physical access to PUF after enrollment. (**b**) demonstrates a sketch for securing exposed CRP by encoding the challenge on the server side and masking the response from the PUF side.

The second mode of communicating with the PUF-enabled device is the mission mode. In this mode, a communication channel is established between the PUF-enabled device and a trusted party. Figure 2b shows a sketch of a secure communication between the Server as the trusted party and the PUF-enabled device. In this mode, it is assumed that the communication channel securely transmits the CRPs while the true values of the CRPs are hidden. For instance, we could assume the challenges are encoded on the server side before being transmitted to the device. An example of challenge encoding is given in [20]. In this work, it is proposed to encode a given challenge value $c$ before transmission, by recurrently shifting the value for $n$ times. At each time, the modified challenge value is fed to the PUF for a new response value $r_i$ where $i \in \{1, 2, \ldots, n\}$. At the end, an extension of the response values with repeating bits is then XORed with the challenge value $c$. This way, the challenge value is encoded to hide its true value during the transmission.

Moreover, we assume that the response values are masked before being transmitted from the device to the Server. We assume that only the device and the server have the mutual data in order to decode and de-mask the challenge and response as received, respectively. Consequently, access from an unknown third party will not yield directly to leaking the CRPs. In such case, additional efforts are required to first break the masked response and also decode correctly the corresponding challenge. Therefore, the cost of preparing a CRP set will increase for a third party with unauthorized access to the communication channel. An example of response masking is discussed in [21]. In this work, the authors propose altering the value of the response bit of the strong PUF according to a trigger value that is yielded from an independent logical composition of some arbitrarily chosen challenge bits. Note that the selection of the challenge bits which constitute the trigger is known on the PUF-enabled device and the Verifier Server.

Additionally, the structure of the strong PUF can play a role in securing the PUF against model building. It is being widely discussed in the literature that model-building of strong PUF is theoretically possible. However, practically, it depends strongly on the level of the PUF structure's complexity [12,13,22]. Therefore, to increase security of the PUF, it can be suggested to employ PUF with increased design complexity. However, for enrollment of the PUF with ML-based techniques, it is then crucial to make sure, on the other hand, that the modelability of the PUF on the server side during the enrollment is facilitated. For instance, it is suggested in the Slender PUF protocol design to build an equivalent model of an XOR Arbiter PUF during the enrollment via having access to each individual Arbiter PUF block. It can be assumed here that, for increased security, the XOR

size of the PUF is large—while, with the eased access to the individual Arbiter PUF, it is still viable to accurately model the XOR PUF for enrollment using a compact CRP dataset.

Including such secure solutions for protocols which use ML-based modeling for PUF enrollment is crucial. It is important to ensure that only a trusted party e.g., a verifier Server, can have an accurate model of the PUF. Therefore, access to the CRPs of the PUF can be open to the trusted party in a trusted zone. While outside of the trusted zone, the openly accessible CRP is disabled and the CRP transmission is secured in a way to hide the true value of the CRPs. In the following section, we elaborate more on what constitutes the estimated model of a strong PUF.

## 4. Machine Learning Modeling of Strong PUF

The goal of strong PUF enrollment with ML-based modeling is to replace the conventional CRP database with an estimated model of the PUF. Let us denote the estimated model of PUF as $h_{PUF}$. The enrollment of PUF with ML-based modeling means that the verifier server will own $h_{PUF}$, which provides access to the full CRP space of the PUF circuit with some miss-prediction error that is tolerable (see Figure 1a). This should in turn mean that the $h_{PUF}$ and the PUF circuit itself should respond similarly to any given challenge from the CRP space. Let us consider $c_i$ as a challenge input to the PUF circuit. If we observe the PUF circuit as a function $f_{PUF}$ of $c_i$, then its estimation can be defined as a function $g_{PUF}$ of $c_i$ and a set of internal values $\theta$ of the model. Thus, $h_{PUF} = \{g_{PUF}, \theta\}$. The estimation should then follow (1):

$$f_{PUF}(c_i) = r_i \approx r_i' = g_{PUF}(c_i, \boldsymbol{\theta}) = h_{PUF}(c_i) \tag{1}$$

where $r_i$ is the PUF circuit's response to the challenge $c_i$ and $r_i'$ is the estimated model's prediction of $r_i$ for $c_i$. The model then goes through an iterative training phase, where a learning algorithm modifies the internal values with respect to the CRP set and the function $g_{PUF}$. At the beginning of the training phase, model $h_{PUF}$ has a significant probability of erroneous estimation of the PUF's CRP characteristic. Therefore, the training runs iteratively until the probability of erroneous estimation is converged to zero or an acceptable minimum value.

Since modeling here is done for the enrollment, we define metrics that are important for the enrollment, and we use them to evaluate the cost of training and the performance of the estimated models:

- **Prediction Accuracy ($\epsilon$):** Proportion of correctly predicted responses to total number of predictions.
- **Enrollment CRP Set Size ($css$):** Size (in bytes) of the CRP set collected to enroll a given PUF circuit.
- **Total Time of Training ($T$):** The time of training in seconds, up to a point when an estimated model is generated with acceptable $\epsilon$.
- **Estimated Model Size ($ms$):** A measure of size (in bytes) of the internal trainable parameters $\theta$ of an estimated model of a PUF.

## 5. Proposed Enrollment Procedure

The PUF enrollment procedure is done by an authorized party with open access to the PUF circuits to collect arbitrary amount of CRPs. We refer to this authorized party as the designer.

Before the enrollment process, we assume that the designer first has performed a CRP-readout on a group of silicon Chips. During the CRP-readout, the designer captures an initial set of CRPs for each strong PUF circuit. We refer to this initial set as $CRP_t$. We assume that the size of the $CRP_t$ is fixed for all the PUF circuits in the same group. The $CRP_t$ is then divided into three subsets. We define a $CRP_{tr}$ subset to be used for training the estimated model. $CRP_{val}$ is a subset to evaluate the estimated model during training, and $CRP_{te}$ a

subset to evaluate the estimated model after training. We assume $CRP_{te}$ to be considerably larger than $CRP_{val}$.

During the process of training the estimated models for the PUF circuits, we take the following considerations as well:

1.  The PUF circuit and its estimated model respond similarly to any randomly given challenge with high probability. In this case, we say that the estimated model has a high value for $\epsilon$.
2.  The number of CRPs needed to train each estimated model is enumerable and feasible to collect.
3.  The training process is finite and the training time $T$ for obtaining an estimated model is minimum.
4.  The estimated model's internal parameter set $\theta$ is enumerable and feasible for storage.
5.  Each estimated mode characterizes only its corresponding PUF circuit, and has no correlation with other PUF circuits in the same group of PUFs.

The procedure of ML-based enrollment can then be developed as shown in Figure 3. This procedure will run in three phases to enroll a given PUF circuit: (1) the Initialization phase, (2) the Optimization phase, and (3) the Evaluation phase.

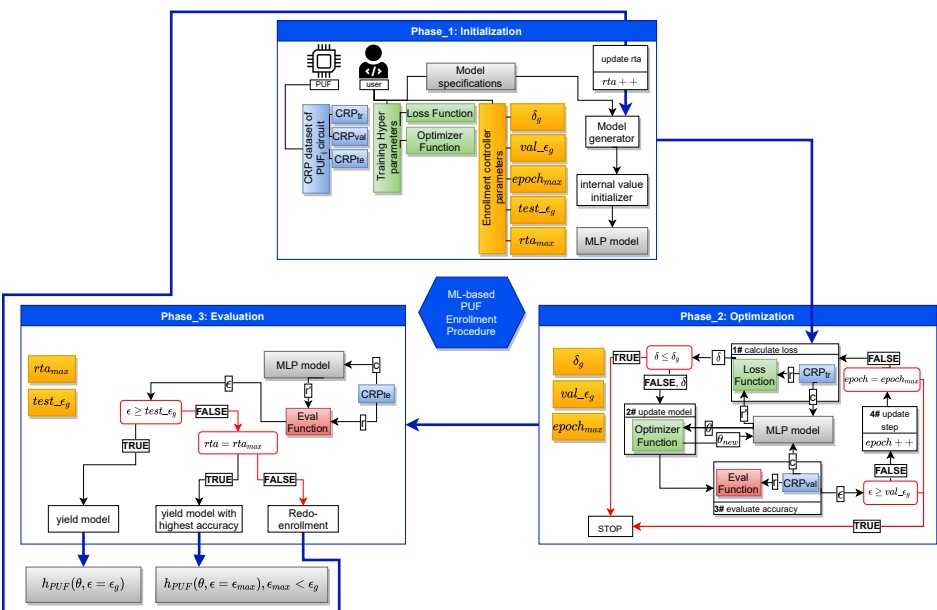

**Figure 3.** Our proposed ML-based enrollment procedure.

### 5.1. Initialization Phase

Here, we define and initialize the necessary parameters for the optimization and the evaluation phase. The control parameters as described in Table 1 are initialized by designer specified values. These parameters define the target accuracy of the estimated model as well as the maximum period of time the enrollment can take to reach the model with the target prediction accuracy. The Training Hyper-parameters are also initialized in this stage. These parameters are machine learning specific and necessary to be assigned according to the context. The choice of the values for these parameters is up to the designer as well, and the best practice is to inherit the values of the successful practices in the literature that have done the strong PUF modeling. The training CRP set $CRP_t$ is also given at this stage, including its three subsets as explained earlier. Finally, the estimated model is created in this stage. The model specifications are also given by the designer. After the model is created, it is sent to an initialization process where its internal values are randomized. The model initialization part of the phase can be iterative. It would depend on how the next two phases will perform, which we explain in the following.

**Table 1.** Control parameters in the enrollment procedure.

| Parameter | Description |
| --- | --- |
| $val\_\epsilon_g$ | Desired value for $\epsilon$ with respect to $CRP_{val}$ subset. |
| $test\_\epsilon_g$ | Desired value for $\epsilon$ with respect to $CRP_{te}$ subset. |
| $\delta_g$ | Desired average error of mis-prediction of the estimated model with respect to the $CRP_{tr}$ subset. |
| $css_{tr}$ | Size (in bytes) of the $CRP_{tr}$ subset. |
| $epoch_{max}$ | The total number of iterations performed during training an estimated model. |
| $rta_{max}$ | Number of times the training can refresh on modeling a given PUF circuit. |

*5.2. Optimization Phase*

The training of the estimated model is done in this phase. It comprises multiple functions for training and evaluation of the estimated model. At the beginning of the phase, the estimated model is given a set of challenges from the $CRP_{tr}$ dataset. The model then predicts the corresponding responses. The loss function will take the predicted responses and the actual responses from the $CRP_{tr}$ and compute the prediction loss $\delta$. Then, the loss value $\delta$ is given to the optimizer function which propagates adjustments to the internal parameters $\theta$ of the estimated model. Then, the Eval function computes the prediction accuracy $\epsilon$ of the updated estimated model, using the $CRP_{val}$ dataset. This entire process is called an epoch, which is counted as *epoch* in the procedure. The procedure undergoes several epochs of training until the desired value of loss $\delta_g$ or prediction accuracy $val\_\epsilon_g$ is observed or *epoch* reaches max value $epoch_{max}$.

*5.3. Evaluation Phase*

The evaluation phase performs the final prediction accuracy assessment of the updated estimated model over the $CRP_{te}$ dataset. Here, the same eval function measures the prediction accuracy of the model over $CRP_{te}$. This evaluation in turn tries to emulate the scenario where the estimated model is invoked during mission mode to communicate with the PUF circuit. In such case, it is justified to have the $CRP_{te}$ set size be considerably larger than that of $CRP_{val}$.

The control sequence in this phase compares the prediction accuracy $\epsilon$ of the model with the $test\_\epsilon_g$. If greater, then the the enrollment yields the model as an estimated model with desired accuracy for enrollment. If the prediction value is less than $test\_\epsilon_g$, however, the control sequence sets to redo the training from the initialization phase where the model's internal parameters are initialized. The number of re-training times is also counted in the procedure with $rta$ counter. If $rta$ reaches $rta_{max}$ as defined by the designer, then the control sequence yields the enrollment with the model with maximum accuracy $\epsilon_{max}$ from the previous training attempts, where $\epsilon_{max} < test\_\epsilon_g$.

**6. The Methodology**

Recall that we assume in this work that enrollment is done for a very large number of PUF circuits. Primarily, the user has to initialize the control parameters before to conduct the enrollment procedure. We know already that the ML-based enrollment procedure is empirical, meaning that there is no deterministic parameter initialization known beforehand to initiate the ML-based enrollment with and yield the desired results. Instead, the optimal values to start with are empirically drawn from a test case. Accordingly, for enrollment of PUF using ML-based modeling, we suggest that the user performs the enrollment in two parts:

- Part 1: Set arbitrary values for the hyper parameters, and desired values for the control parameters $val\_\epsilon_g$, $test\_\epsilon_g$ and $\delta_g$, and maximum tolerable values for $epoch_{max}$ and $rta_{max}$. In addition, define a range of different $css$ values. Then, perform the

enrollment procedure over each *css* separately and evaluate at which *css* the desired $\epsilon$ is reachable with acceptable *T*.

- Part 2: Update the control parameters with implications of the optimal values for $\epsilon$, *T* and *css*, obtained from part 1, and resume the enrollment on the rest of the PUF circuits with the updated control parameters.

Our focus here is mainly on part 1. Thus, our goal is to show how the observations obtained in Part1 can help the user to define the optimal values for the control parameters. Speculatively, to be cost-efficient for the enrollment of a given large group of PUF, the user is able to find a minimum value for *css* given the implications he receives from the evaluation on the primary subset in part 1. Therefore, he will be able to avoid larger *css* and save time during the CRP-readout for the remaining large number of PUF circuits pending for enrollment.

Since it is an empirical method, we should first define the hard bounds for either one of the control parameters. We suggest to do this for the most critical parameter such as $css_{tr}$. After exploring the values within the hard bounds, then find the optimal points for the cost values. For instance, looking at where *css* or *T* is minimum, the $\epsilon$ is maximum and *css* is minimum. Accordingly then, update the values of the control parameters (see Figure 4).

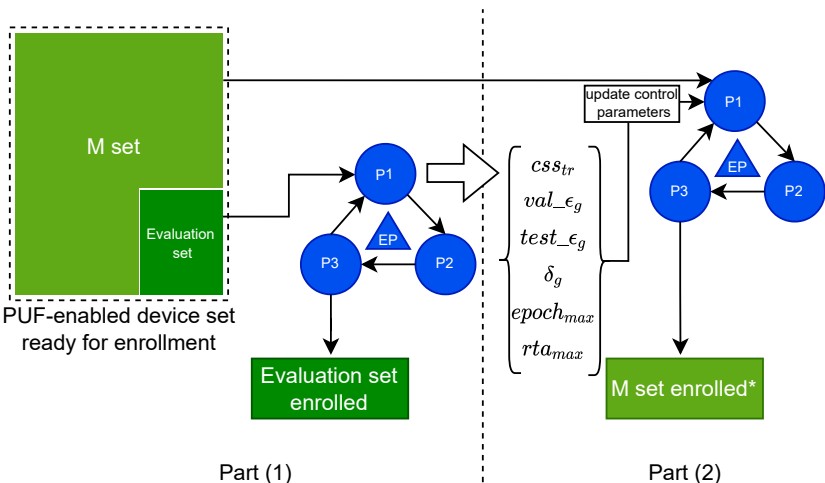

**Figure 4.** Our proposed enrollment method. Here, EP refers to our proposed enrollment procedure. M set also refers to the main set of PUF devices to be enrolled with adapted control parameters.

Additionally, an exploration of *ms* can be done. *ms*, however, is mostly relying on the parameters which constitute the structure of the probabilistic model, such as number of neurons and weighted connections for each neuron if the model is an Artificial Neural Network (ANN). *ms* parameters are quite numerous. Therefore, the exploration over *ms* should be selective, such as exploring different model structures that have already been proposed in the literature. The method of grid searching the hyper-parameters for training also exists, such as the learning-rate value, the optimization function, etc. This can be done on the side of the enrollment method as we define here.

Once the enrollment on the evaluation set is complete after exploring different values for the control Parameters, there will be two products: one which is the enrollment of the devices in the evaluation set, and second is the optimized control parameters which then can be used in the enrollment of the *M* set (see Figure 4). This will constitute the second part of the method. Contrary to the first part, in the second part, no bound for control parameters are set. Instead, final values for these parameters are given, which are coming from the first part. The outcome of the second part of the method is to accept an $h_{PUF}$ model which meets the qualifications according to $\epsilon$ parameters, or discard the model which does not have the desired quality. In case of discarding the models, their corresponding PUF will be queried again for more CRP-readout. Nonetheless, we speculate that, with the

optimized control parameters exported from part one, the population of discarded models in the second part should be minimized considerably.

## 7. Evaluation Work

In this section, we show in an experiment how we analyze *css*, *T* and $\epsilon$ over a small batch of 100 2-XOR Arbiter PUFs.

### 7.1. Specifications

We considered using XOR Arbiter PUF as our target PUF model family in our evaluation code. We conduct our experiments on data generated from a Python based Arbiter PUF and XOR Arbiter PUF simulation. We elaborate on XOR Arbiter PUF and its simulation in Python in the following.

XOR Arbiter PUF is considered as a variant of the Arbiter PUF family. Arbiter PUF was first introduced in 2002 by Gassend et al. in [23]. The idea of Arbiter PUF is based on the delay difference between two racing paths that are structurally similar, but, due to minor process variations, they differ in time of passing a signal given to them at the same time. The structure of XOR Arbiter PUF is based on multiple Arbiter PUFs whose input (challenge) is of the same size, and triggered by a global input. The output of the XOR Arbiter PUF is also the XOR of the output of each Arbiter PUF it comprises. Figure 5 shows the structure of an *n*-stage *k*-XOR Arbiter PUF.

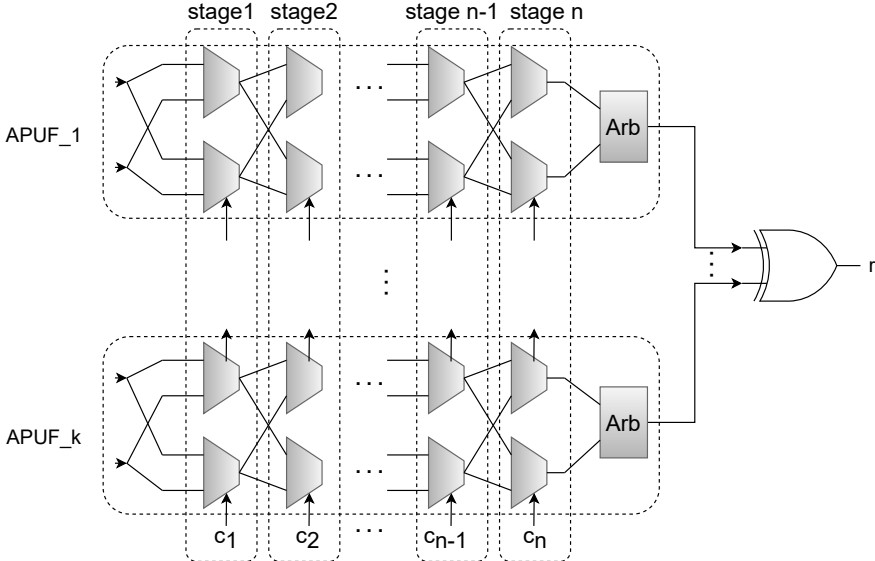

**Figure 5.** Illustration showing the structure of n-Stage k-XOR Arbiter PUF.

We reused the XOR Arbiter PUF simulator developed by Ruhrmair as described in [24]. The source code of this simulator can also be found in [25]. In this simulator, the two racing signals' propagation delay is modeled as the sum of the delays in each stage. The delay parameter values in the Python-based implementation of APUF and XOR Arbiter PUF simulator are generated randomly according to a standard normal distribution, with mean 0 and standard deviation 1.

For our experiment, we generated 100 instances of 128-stage 2-XOR Arbiter PUF. We then randomly generated 35,000 challenges for each instance, and recorded their corresponding response. Thus, we stored 100 CRP datasets with 35,000 CRPs in each set. Note that the datasets generated from the simulated PUF instances do not simulate the instability that is inevitably present in real PUFs. We intentionally chose the instability-free condition, since the presence of instability is a new fold of complexity that can affect the modeling results, and thus it needs to be discussed thoroughly in a separate set of experimental work.

To assure the reliability of the simulated instances and the generated CRPs, we measured the randomness, uniqueness, and diffuseness, using the formulations proposed

in [26,27]. A collective use-case of Hori's Uniqueness and Maiti's Uniqueness, as well as the randomness and the diffuseness, can be found in [14]. Our measurements of uniqueness, randomness, and diffuseness over 100 instances of 128-bit 2-XOR Arbiter PUF are brought in Table 2. Note that each CRP set considered for these measurements comprises 10,000 CRPs.

**Table 2.** Measurements on 100 PUF instances CRP sets.

| Average Randomness | Maiti's Uniqueness | Hori's Uniqueness | Average Diffuseness |
|---|---|---|---|
| 0.9419 | 0.4999 | 0.9899 | 0.9972 |

We chose our estimated model to be Multi-Layer Perceptron (MLP), which is a variant of Artificial Neural Network (ANN) models. Mursi et al. in [15] has proposed a structural definition of MLP model for modeling XOR Arbiter PUFs that has the potential to converge faster with a considerably lower number of CRPs for training compared to other modeling structures such as ones discussed in [24,28], which are based on Logistic Regression (LR), and Ref. [29], which is based on Artificial Neural Networks (ANN). A schematic of Mursi's proposed MLP structure is given in Figure 6. Here, $k$ is with the number of XORs in an $n$-stage $k$-XOR Arbiter PUF. The mentioned feature vector in the figure is solely a function of the applied $n$-bit challenge $c$. As also described by Ruhrmair in [24], the feature vector as $\vec{\Phi}$ can be defined as $\vec{\Phi}(c) = \prod_{i=1}^{k}(1 - 2b_i)$, where $b_i$ is the $i$th bit of the challenge $c$.

Accordingly, for a 128-stage 2-XOR Arbiter PUF, the MLP model we use in this work has an input layer with 129 neurons, first hidden layer with two neurons, second hidden layer with four, and a third layer with two neurons. The output would also have one classifier neuron. For the training hyper-parameters and the enrollment control parameters, we considered the values given in Table 3. Here, the parameters marked with * are the control parameters of the enrollment.

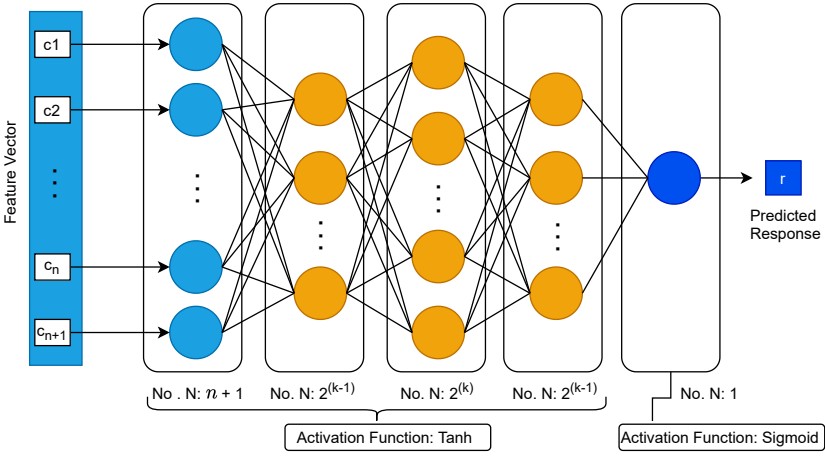

**Figure 6.** Illustrating the MLP structure proposed by Mursi et al. in [15].

**Table 3.** Hyper-parameters set for the initialization phase.

| Parameter | Optimizer Function | Loss Function | Learning Rate | Weight Initializer | Bias Initializer | * $epoch_{max}$ | * $test\_\epsilon_g$ | * $rta_{max}$ | * $val\_\epsilon_g$ | $\delta_g$ | CRP_{te} Set Size |
|---|---|---|---|---|---|---|---|---|---|---|---|
| Value | Adam | BCELoss | 0.001 | Taiming Uniform | Uniform | 400 | 10 | 90% | 99% | 0.01 | 20,000 |

### 7.2. Experimental Observations

We did an exploration to find the ideal values for $css_{tr}$ and $T$. The choice of the lower bound of $css_{tr}$ and the upper bound was also arbitrary. We inferred from the previous studies that the lowest value for $css_{tr}$ for a 2-XOR 128-it XOR Arbiter PUF is a value of

about 3000 CRPs. Therefore, we choose that as the lower bound. For the upper bound, we chose 10,000 CRPs. We speculated that the characteristic we observe around this number of CRPs can be interpolated for larger sizes of $css_{tr}$ as well.

We used Pytorch in Python 3.7 to build and train our ANNs. We conducted our experiments on a PC running windows 10 with an Intel core i7 8th Gen CPU and 16 GB of memory. We developed our experimental Python codes using Spyder 4.0.1 on Anaconda Navigator 1.9.12.

The results on the performance of training with various training set sizes are reflected in Figure 7. Looking at the results, we can identify at what $css_{tr}$ value the probability of reaching the desired $\epsilon$ is very low, which is between 3000 to 4000. Note that the model accuracy at these $css_{tr}$ values is $\epsilon_{max}$ as we indicated in the procedure. In addition, looking at the range between 4500 CRPs to 10,000 CRPs for $css_{tr}$, it infers that, with increasing the set size, the probability of reaching the desired $\epsilon$ tends to stabilize at a value above the target $test\_\epsilon_g$ as we defined in the test phase. There exist some outliers that are cases where the desired $\epsilon$ could not be reached; therefore, the model with $\epsilon_{max}$ is yielded. This can characterize the possibility of having discarded $H_{PUF}$ models during Part 2 of the enrollment method as we explained earlier.

Moreover, the training time $T$ tends to decrease as well at the beginning with increasing the set size up to 7500. We imply that this reduction in time of training is due to a significant decrease in re-training attempts until desired model is yielded.

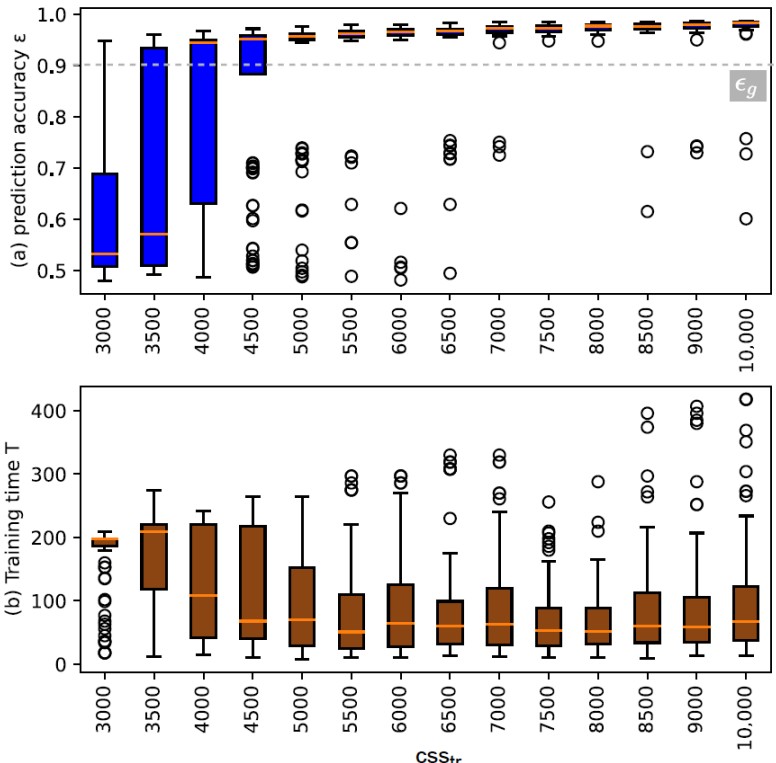

**Figure 7.** Illustrating the distribution of prediction accuracy $\epsilon$ and the distribution of total training time $T$ over various $css_{tr}$.

Plots shown in Figure 8 also show the minimum, maximum, and average values of $T$ and $\epsilon$. Looking at Figure 8a, we can observe what are safe values of $css_{tr}$ in terms of delivering the target prediction accuracy $\epsilon$ with maximum probability. In this scenario, look for $\epsilon > 0.90$, that is, the values between 7000 to 8000 CRPs. Looking at values of $\epsilon$ and $T$ shown in Figure 8c, one can infer at what $css_{tr}$ values there is a chance of obtaining the target $\epsilon$. Here, at 3000 for instance, 3000 CRPs could yield a model with $\epsilon > 0.90$. This, however, means that a designer needs to improve other factors to increase the chance of

obtaining model with target $\epsilon$. For instance, by using a better initialization technique or modifying the model structure, which could in turn affect the model size *ms* consequently. By also looking at Figure 8b, we can infer at what $css_{tr}$ values we have an increased chance of obtaining the target $\epsilon$. For instance, for $css_{tr} > 5000$, it is possible to obtain prediction accuracy $\epsilon > 0.90$. However, for this $css_{tr}$, there is a considerable chance that models with $\epsilon < 0.90$ are yielded (see Figure 7). Choosing this option depends on the cost of re-querying the corresponding PUFs for the outlier predictive models with low prediction accuracy. If the cost of re-querying is amenable by choosing the low $css_{tr}$ for the majority of the PUF devices for the first query, then it could be considered a potential choice in terms of lowering the overall cost of enrollment.

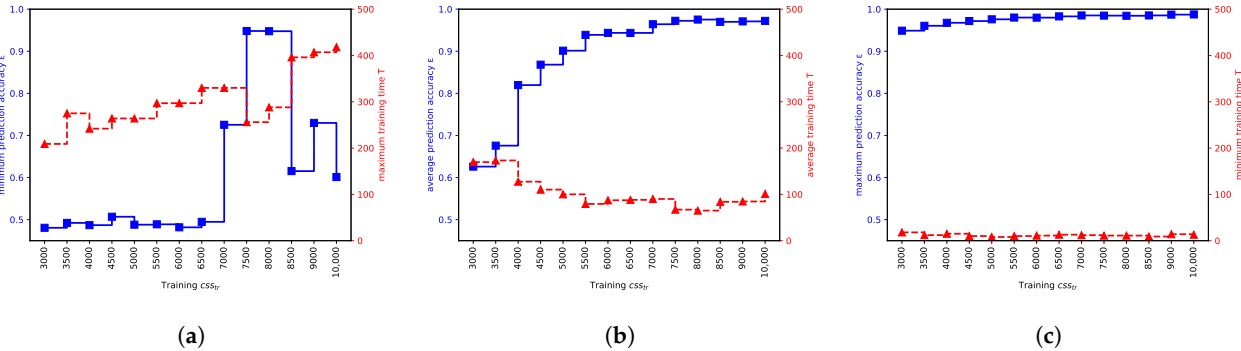

(a)         (b)         (c)

**Figure 8.** Illustration showing the minimum, average, and maximum training time *T* and prediction accuracy $\epsilon$ for various $cc_{tr}$. (**a**) Max *T* and Min $\epsilon$; (**b**) average *T* and $\epsilon$; (**c**) Min *T* and Max $\epsilon$.

We could have two approaches here: (1) To choose the minimum *css* that has a chance to yield some $h_{PUF}$ model with sub-optimal $\epsilon$; (2) To choose *css* which yields maximum $\epsilon$ and has a negligible chance of yielding $h_{PUF}$ with sub-optimal $\epsilon$.

Choosing the first option can yield in overall reducing the CRP read-out cost since we chose the minimum $css_{tr}$. However, since there is also the chance of yielding $h_{PUF}$ with sub-optimal $\epsilon$, then there may be some additive cost of re-querying the PUFs with discarded $h_{PUF}$ for re-enrollment. If the cost of re-querying is tolerable, then the first option could potentially be the cost-efficient choice. Choosing the second option, however, could yield the overall increased CRP read-out cost. We also saw that training time *T* can also be minimized if the prediction accuracy $\epsilon$ is maximized with increasing *css*. However, the chance of yielding $h_{PUF}$ with sub-optimal $\epsilon$ is negligible on the other hand. This option could be a potential choice for cost-efficiency if the cost of re-querying the PUF is high.

We also measured the size of the trained estimated models. The MLP used in this study comprises 282 trainable parameters in total. Each parameter is 32-bit floating point, yielding in total a model size of 1128 bytes. Note that we do not need to save any metadata regarding the internal connectivity of the MLP, since all layers in the MLP model are Fully-Connected. While we cannot make a thorough comparison at this level of evaluating the method, we can at least draw a primary conclusion that storing an estimated model of a strong PUF circuit potentially takes much less space, compared to that of a CRP set. For instance, the training CRP set used here to obtain the most accurate model includes 10,000 CRPs, equal to 161 KB of storage size, whereas the estimated model size is roughly above 1 KB.

The uniqueness of the trained estimated models is also an important characteristic, which means that no estimated model should respond similarly to two different PUFs. Although this is not a metric related to the cost of enrollment, it is, however, essential to ensure that models correspond only to their equivalent PUF circuit. This should satisfy the fifth consideration we discussed in Section 5. We refer to it as measuring the uniqueness of the estimated models. Uniqueness here is observed as the prediction accuracy of each trained estimated model over a given CRP dataset coming from each PUF circuit. Since we have 100 PUF circuits and their corresponding 100 estimated models, we therefore mea-

sured 10,000 cases for the uniqueness. We considered two cases, one which is training the models with $css_{tr}$ = 3000, and one with $css_{tr}$ = 8000 CRPs. The results of this measurement are brought in Figure 9. We expect for $css_{tr}$ = 3000 that the models have low similarity to their corresponding PUF. Looking at Figure 9a, we see that only a selective number of cases have $\epsilon$ above 0.70. Nonetheless, for all cases, it is apparent that no similarity with $\epsilon$ higher than 0.6 is achieved. We also observe on Figure 9b that estimated models trained with $css_{tr}$ = 8000 show similarity uniquely only to their corresponding PUF circuit with high accuracy. We cannot infer directly from these observations that, for any trained model on a PUF circuit in general, the model shows CRP similarity only to the corresponding PUF. Since we suspect that there might exist some PUF designs that intentionally have CRP similarities scattered between various PUF devices, we can nonetheless infer that, for a set of PUF devices for which their PUF characteristic shows good random and unique behavior (see Table 2), predictive models trained using enough CRPs from a PUF represent uniquely that PUF only, while having no similarity to other PUF instances.

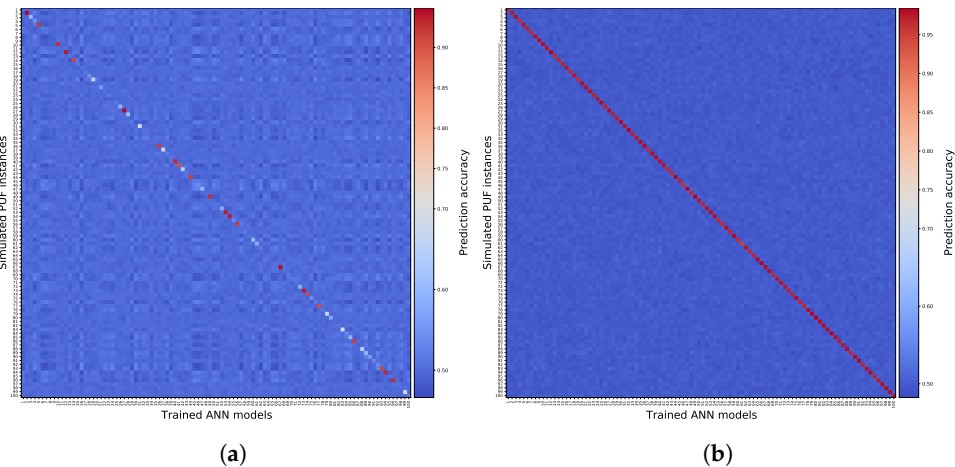

**(a)**                                                 **(b)**

**Figure 9.** Similarity matrices showing the similarity between 100 trained models and 100 instances of a 128-stage 2-XOR Arbiter PUF. (**a**) $css_{tr}$ = 3000; (**b**) $css_{tr}$ = 8000.

## 8. Applicability in the Related Works

The methodology we developed here is applicable to recent trending use-cases of PUF modeling, which are device authentication and encryption key generation. Below, we discuss several PUF based protocols that utilize an equivalent software program of the model of the PUF.

The Slender PUF protocol proposed by Majzoobi et al. in [7] is an authentication protocol that uses strong PUF. In this protocol, the authors propose a substring matching mechanism, wherein the substring in the protocol is a sliced vector from a vector of generated responses both on the PUF device and the verifier server. Authors in this work assume that the verifier server has access to a compact model of the PUF, which is able to generate a response for any given challenge vector, similar to that in the original PUF circuit. The sliced vector of responses is sent from the PUF device to the verifier, and convoluted over the entire response vector that has been generated on the verifier server, using the compact model of the PUF. The authentication is successful once the sliced vector shows maximum correlation to a subpart of the verifier's response vector. Since, in this protocol, the sliced response vector is exchanged on a public channel, it is obliged that, once the vector is used, it is discarded and never used again to prevent replay attacks. This in turn requires that the verifier server has access to a very large amount of the CRP space of the PUF, which is in turn guaranteed by using the model of the PUF. Thus, the model of the PUF should correspond as accurately as possible to the PUF circuit in order to suffice the requirement.

Another novel mutual authentication protocol has been proposed by Idriss et al. in [17,18], which is based on a challenge–challenge communication mechanism between

the PUF-enabled device and the verifier server. It is also assumed in this work that the verifier server acquires an accurate model of the PUF for enrollment. During the mutual authentication, after exchanging the device IDs, the PUF-enabled device generates several random CRPs. To authenticate the server, the challenge vectors of the generated CRPs are sent from the device to the verifier server. On the server side, for each received challenge vector $c_i$, two new random challenge vectors $c_j$ and $c_k$ are generated, such that the XOR of the response values corresponding to each of the two newly generated challenge is equal to the response of the received challenge vector. In other words, the $A(c_j) \oplus A(c_k) = r' == r = A(c_i)$ should hold true, where $A$ is the model of the PUF circuit. Once the $c_j$ and $c_k$ are generated for every $c_i$ received on the verifier server, they are sent to the PUF-enabled device. On the PUF-enabled device, it is then checked to see if the equation $PUF(c_j) \oplus PUF(c_k) = r' == r = PUF(c_i)$ is true for the majority of the received challenge vector pairs. Once the device authenticates the server, it sends a new challenge vector pair set (the same as the verifier did) to the verifier server for authenticating the device. Since this mutual authentication method is based on random generation of the challenge values for every authentication request, it is assumed that the verifier server has access to a large CRP space where, for every randomly generated challenge vector, a response value can be provided. This of course is guaranteed by using the equivalent model of the PUF, which is highly accurately trained.

A PUF based key generation protocol has been proposed by Quadir et al. in [19], which uses a machine learning generated predictive model of the PUF on the TTP server for mutual key generation. Here, the authors propose mutual key generation by exchanging only a serial number, which in turn is the challenge to the PUF device and the predictive model, respectively. It is expected of course that both the model and the PUF device generate the same response value. The response value of course is prone to variations due to device instability and model miss-prediction, which is why the authors also propose using helper data and error correction codes to recover the original key generated on the TTP server for the PUF device. Similar to the idea of PUF authenticating in [18], here only the challenge values are exchanged and no responses, in order to avoid model-building attacks. The protocol proposed here also refreshes the key after a certain period. This feature of course needs both the device and server to have access to a large CRP space, which is again for the verifier server, provided using the model of the PUF. This way, the users will be able to refresh keys frequently and each time guarantee that a new value is generated.

The advent of such protocols for device authentication and key generation enables a secure implementation of One-Time-Password (OTP) methods to be more feasible and reliable than before. Since the CRP space of strong PUF is considerably large, it can be easily guaranteed that every newly generated password is unique. On the other hand, the reliability of the key is of importance, which is partially assured by providing a highly accurate model of the PUF. Additionally, once these protocols emerge into a large spectrum of embedded systems and connected devices, it is expected that the enrollment process now relying on machine learning-based modeling can be practiced for a very large volume of devices. Therefore, the cost of enrollment, as we defined its constituent terms in this work, finds their importance and needs to be managed properly.

## 9. Conclusions

In this paper, we presented a formalized ML-based enrollment procedure for strong PUFs and a two-part methodology to evaluate the cost and performance of training with respect to several metrics. We showed that the evaluation, which happens during the first part of the enrollment method, assesses the cost of enrollment for some given control parameters by the designer. This evaluation then highlights the optimal values for several training parameters such as the training CRP set size, time of training, and model prediction accuracy. We then discussed that the second part of the method uses the optimal training parameter values found in part one, in order to enroll the large group of PUF circuits.

## 10. Future Work

Here, we only evaluated the time of training $T$ and prediction accuracy $\epsilon$ with respect to $css_{tr}$ and model size $ms$. In fact, since no rival ML-based methods have been studied here, we could not compare different values of $ems$ with respect to $css_{tr}$. One of the future extensions of this work therefore is to include exploration on various modeling techniques and investigate optimal suggestions considering also the value of $ems$. An exploration on other MLP model structures can provide that variety for $ems$. However, we cannot solely compare the results with previous works. Such assessments require reproducing the existing estimated modeling techniques and observing their performance on a reproduction quality. Assessment of the machine learning-based strong PUF enrollment with the presence of PUF instability is also another extension to this work in the future. This aspect advances through a new complexity fold regarding the viability of modeling of strong PUF. Additionally, methods aiming at better initializing the estimated model before training are potential extensions to this work. For instance, investigating Transfer learning [30] seems to be a promising approach as it shows merits in the field of machine learning, and can be exploited as well for strong PUF modeling.

Access to the full CRP space using a predictive model of PUF can potentially emerge into new protocols that exploit the abundance of the CRPs. As we discussed, several methods exist that already proposed a use-case of a model of the PUF to perform mutual authentication between a TTP verifier server and a PUF-enabled device [7,17–19]. Given that we assume commonly that the entropy of the PUF output is low, (e.g., $r_i \in \{0, 1\}$ for a PUF with binary response output), it would be possible to incorporate novel correction codes that are capable of locally regenerating a mutual value using the PUF responses. In a future extension of this work, we elaborate on a novel centralized key generation technique that exploits the abundance of accessible CRPs on a TTP verifier to build robust mutual secret key values. We explain there that the robustness is guaranteed on the TTP server thanks to the existence of a predictive model of a strong PUF that is accurately trained, using the same principles we explained here.

**Author Contributions:** Conceptualization, A.A.-P., D.H., V.B. and G.D.N.; methodology, A.A.-P., D.H., V.B. and G.D.N.; software, A.A.-P.; validation, A.A.-P., D.H., V.B. and G.D.N.; formal analysis, D.H., V.B. and G.D.N.; investigation, A.A.-P., D.H.; resources, A.A.-P., D.H. and V.B.; data curation, A.A.-P.; writing—original draft preparation, A.A.-P.; writing—review and editing, A.A.-P., D.H., V.B. and G.D.N.; visualization, A.A.-P.; supervision, D.H., V.B. and G.D.N.; project administration, D.H.; funding acquisition, D.H. All authors have read and agreed to the published version of the manuscript.

**Funding:** This material is based upon the work supported by the French National Research Agency in the framework of the "Investissements d'avenir" program (ANR-15-IDEX-02).

**Conflicts of Interest:** The authors declare no conflict of interest.

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
