# Peer review of "Strong PUF Enrollment with Machine Learning: A Methodical Approach"

_electronics, doi:10.3390/electronics11040653_

Round 1
Reviewer 1 Report
Summary: This paper presents a methodological framework for constructing ML-based PUF enrollment procedures.
Strengths:
- The overall methodology makes sense.
Weaknesses:
- This work lacks novelty. The ML-based PUF enrollment has been proposed in many past papers, and the overall methodology is very straightforward. It is quite natural for one to construct a framework like the one presented in the paper. I did not find anything that could justify the necessity for publishing this work.
- It would be interesting if the authors could provide a concrete roadmap for quickly finding the optimal parameters.
Editorial comments:
- page 8, line 265, "pending pending"
- page 7, line 238 "a estimated"
- page 2, line 91, "the the"
Author Response
We thank the reviewer for their time and insights given in the comments. Written below also, we tried to answer some of the comments (The comments are in bold text):
- This work lacks novelty. The ML-based PUF enrollment has been proposed in many past papers, and the overall methodology is very straightforward. It is quite natural for one to construct a framework like the one presented in the paper. I did not find anything that could justify the necessity for publishing this work.
Regarding the lack of novelty as mentioned in the first comment: to the best of our knowledge, the first time that a research work elaborately discusses strong PUF modeling for the purpose of PUF enrollment.
We acknowledge, as we did also in our paper (see the introduction, paragraph starting on line 57, page 2) that there exists a large number of research works that discuss PUF modeling as an attack model. The research in this sector is almost mature.
However, there is another sector in researching PUF modeling, which elaborates the utilization of PUF models in novel authentication and encryption protocols. However, none of the research works in this field discuss elaborately the enrollment methodology. We added a new section to the manuscript which elaborates on several protocols that use a predictive model of PUF for authentication and key generation. This added content should help to clarify where our work is applicable to (See section 8).
The difference between the first sector and the second one is that in the first sector, the maximum effort in modeling PUF is given to a single PUF target. While for the second sector, the modeling SHOULD be discussed for a large group of devices (millions or several million units in manufacturing orders). However, no works have discussed whether there exists an optimal modeling solution for a large number of PUF units to serve the enrollment job. Most of the works in the second sector also, take into account by default that a predictive model of a PUF is already given.
The necessity of publishing our work is justified for the second sector. As the reviewer may have noticed from reading our manuscript, we attempted to compile a method for strong PUF enrollment based on machine learning which encapsulates the important metrics in PUF modeling which are in turn useful to assess and optimize the modeling procedure for a large group of PUFs. Such overview on PUF modeling is lacking in the literature to the best of our knowledge, which in turn makes our work a needed contribution to the literature. We emphasize that our work will in turn be the base-ground for optimization techniques on PUF modeling to further decrease the cost of creating the predictive models ready for enrollment. Taking into consideration that at the current stage, while such work is lacking to refer to in the literature, it would be difficult to discuss optimization techniques in PUF modeling which would serve in enrolling the PUF.
- It would be interesting if the authors could provide a concrete roadmap for quickly finding the optimal parameters.
Regarding the second comment, we emphasize that we already have provided as part of our methodology, an empirical measurement of an evaluative training job, which provides optimal training parameters for a large group of PUF units. This is essentially a practice which leads to yielding the predictive models of the PUF units in the fastest possible time, requiring the least number of CRPs as we explained already in the paper.
- Editorial comments:
The editorial comments have been addressed and fixed also in the revised manuscript.
Noting that in the revised manuscript, the edited and added contents are marked with green text. We did this to make the newly added parts easy to follow. We will of course return them back to the default color for the final version of the manuscript.
We thank again the reviewers for their comments and appreciate their time and effort.
Reviewer 2 Report
The authors have proposed a design idea here where ML model parameters used to estimate a strong PUF will be stored instead of the PUF CRP themselves, saving the space required to the store CRP. Using existing and highly accurate ML/deep learning model algorithm, the authors have shown how it can save space for 100 different simulated PUF instances. The idea to use PUF modeling in this way is novel to the best of authors' knowledge. However, I have some questions for this manuscript that I'd like to have more clarification on:
- For device side, adding an approximate model instead of fully accurate model should increase the complexity of the error correcting codes and consequently increasing computing time/storage which should be taken into account.
- The model used to estimate the PUF response should be fully described as it was not clear how the python model takes the different number of challenge parameters into account and also if it is indeed modeled by an arbiter PUF where each challenge bit compares two closely related delays from a pair of electrical paths.
- The authors have used a fully trained model with high accuracy. It'd be interesting to see a study where if reducing the accuracy might help converge for new data and for new devices.
- A correlation study of this approximate model for different devices would be an interesting read too since we don't want one model to be able to predict responses for another PUF.
- It is difficult to see where this application will be implemented. For server side, storing tens/hundreds kb of data vs 1kb data shouldn't be that big of an issue because the space required to store the CRPs of even millions of devices is not difficult with modern high storage servers. Since this is the main motivation of this manuscript, I'd suggest authors to find such applications areas where storing CRPs can be prohibitive.
Author Response
We thank the reviewers for their helpful insight and their comments.
In answer to the concerns given in the comments, we prepared the following responses (Comments are in bold text):
1- For device side, adding an approximate model instead of fully accurate model should increase the complexity of the error correcting codes and consequently increasing computing time/storage which should be taken into account.
We indeed accept that the miss-prediction error of the predictive models, even though marginal, can affect the communication between the verifier server which houses the predictive models, and the PUF-enabled device. Nevertheless, this depends heavily on the application and the protocol restrictions and constraints. Since we annotate in the manuscript that highly accurate models should be provided, then we can justify that the error in predicting the response induced from the miss-predictive characteristic of the model, can be corrected with existing error correction codes, the same as it is possible for the PUF source. The marginalized erroneous responses coming from the model can be tolerated since we expect above 99% accuracy in the response prediction. For instance, in a PUF authentication scenario, if we expect to send 100 CRPs from device to the verifier server, assuming that the PUF source on the device has 5% chance of response bit-flip, then for 100 single-bit response CRPs sent from the device to the server, in the worst case scenario, 95 responses will be correct. Given that the predictive model on the other side have 99% percent accuracy, thus for the received CRPs, in the worst case, the model predicts 99 response correctly out of 100. This will give us in total, a 94 matching CRPs assuming that the miss-prediction and the CRPs with flipped response does not overlap. At such rate, using majority voting for instance, we can still give a confident answer for device authentication.
Nonetheless, we acknowledge that 100 CRP generation and transmission, may not be easy to convey for all devices. That is why we indicated initially that it is heavily depended on the protocol restrictions and constraints. If a protocol for instance allows only a few CRP transmission per authentication request, then the model accuracy and the reliability of CRPs coming from the PUF source should be amended using local reliable CRP extraction techniques.
We added a new section (Section 8) to the manuscript which elaborates on several protocols that use a predictive model of PUF for authentication and key generation. This section should give a brief insight into the applicability of the predictive models into applications such as device authentication and encryption key generation.
In an extended version of our work, we are elaborating on novel communication methodology that exploits the full CRP access using the predictive models, to build robust and secure mutual-authentication and key generation method. We indicated our future approach in the revised manuscript the future perspectives section of this paper.
2- The model used to estimate the PUF response should be fully described as it was not clear how the python model takes the different number of challenge parameters into account and also if it is indeed modeled by an arbiter PUF where each challenge bit compares two closely related delays from a pair of electrical paths.
We emphasize that we already tried to give all the details about the predictive model structure we employed in our case study. Information given on Section 7.1 (including Table 2 and Figure 6, and textual content starting from line 355) elaborates in details the parameter values we considered in constructing the MLP model and the training parameters. We also explained how the Challenge vector is transformed into a feature vector, which is the input the the predictive MLP model (Line 364). We also referenced a previous work which uses the feature vector representation of the challenge vector (Reference number 25).
3- The authors have used a fully trained model with high accuracy. It'd be interesting to see a study where if reducing the accuracy might help converge for new data and for new devices.
4- A correlation study of this approximate model for different devices would be an interesting read too since we don't want one model to be able to predict responses for another PUF.
The point given in this comment is important, and we elaborate on this measurement in terms of model prediction uniqueness, which has already been included already in our paper (Starting from Line 435, and see Figure 9 as well). We agree that measuring the correlation of predictive models to other PUF devices is important as it can be a vulnerability against impersonation attacks. We can include an added measurement which shows the correlation of the predictive models not with high prediction accuracy, to their corresponding and non-corresponding PUF devices (See Figure 9(a)). This should also suffice the requirement imposed on comment 3.
5- It is difficult to see where this application will be implemented. For server side, storing tens/hundreds kb of data vs 1kb data shouldn't be that big of an issue because the space required to store the CRPs of even millions of devices is not difficult with modern high storage servers. Since this is the main motivation of this manuscript, I'd suggest authors to find such applications areas where storing CRPs can be prohibitive.
We agree that a detailed description of the application of ML models of PUF in this manuscript is lacking. However, we would like to clarify that including such material into this work, demands a large extension, which may take away the attention we want the readers to have on the importance of the enrollment process. We however follow this comment and add an extended discussion which links our work to existing protocols and PUF based communication methods which operate using estimated models of PUF on the verifier server. The newly added section 8 which elaborates on several protocols that use a predictive model of PUF for authentication and key generation, should help to clarify where our work is applicable to (See section 8).
Noting that in the revised manuscript, the edited and added contents are marked with green text. We did this to make the newly added parts easy to follow. We will of course return them back to the default color for the final version of the manuscript.
Reviewer 3 Report
Please see attached file for my report.

Author Response
First of all, we thank the reviewer for reviewing our work and giving insightful comments. We also provide our responses to the concerns mentioned in the reviews:
1- Different types of PUFs, have been discussed in the literature. The authors should state explicitly that not all PUFs are strong, and they should give specific examples of strong PUFs, with the appropriate references.
We agree that an explanation on the categorization of PUF variants can be added to the manuscript. Following this comment, we added a description to the introduction section, describing the weak and strong PUF variants and their sub-models (starting from 24 to line 37, page 1).
2- A reference should be given for the multi-PUF scenario discussed in lines 32-38.
We noted this comment and added 2 new references to previous works that implement lightweight PUF with multi-bit response output.
4- Line 178: Please explain the notation CRP_t.
Explanation added for CRPt.
5- What are the units of time in Figs. 7 and 8?
We measured the time in seconds, as we also explained in the textual content.
6- In Fig. 8, the authors plot the average prediction accuracy. It seems to me that the minimum prediction accuracy is more important, because it refers to the worst-case scenario
We agree that the choosing the optimal values for ε and T and css by observing the minimum accuracy can be in turn more reliable. But we note that outlier cases exist even in csstr values which in majority yield highly accurate models. And these outliers will in turn represent the minimum accuracy (See Figure. 7), which in turn will allow the designer to overlook the potential csstr value.
However, we think that this could in turn be a good comparison case, as to show how different observations on the ε and T values with respect to different csstr can change the decision baseline of the designers in order to graduate the optimal value choices for the training and enrollment parameters. In the revised manuscript, we added two additional plots to Figure 8, which show the Maximum T and Minimum ε , and Minimum T and Maximum ε together, respectively. We also discuss and compare the different observations on these plots in part of the revised manuscript (starting from line 396 to 412).
7- Let us assume that a token has been enrolled by means of the ML-based enrollment discussed in the present work. At a later stage, the owner of the token, has to be authenticated to the system. How long will the verification take? Milliseconds, seconds, or minutes?
We would argue with this comment, given if we understood correctly that the reviewer meant by verification, the process of verifying an authentic device during mission mode (which is a post-enrollment operation mode). The verification in fact should happen during several nano-seconds. However, we clarify that the mission mode is not the mode we are focused on in this research. Moreover, the verification time measurement is a matter that strongly relates to, first the protocol restriction and constraints, and second the computation power of the PUF-enabled device. In a future work which follows this paper, we are elaborating on a novel protocol. This work has a similar measurement on which we assess the computation overhead of our protocol. However for this paper, since we are focused on the enrollment mode, such measurement is not necessary as we expect.
8- ML-based enrollment is proposed as a means to reduce the amount of data that are stored in the database. Does the ML algorithm require the storage of data? If yes, how much do we gain?
Yes, the ML algorithm still requires storage of data. This data is in fact the weight and bias values of the trained model, given that the model structure is an Artificial neural network. For other ML model structures also, such as Logistic Regression, the trained coefficient vector is the data that needs to be stored. In fact we already discussed this in our paper, following the discussion on line 391 to 400. As we explain there, the storage gain in turn is relative to what PUF structure we are modeling, and what is the initial application's need in terms of number of CRPs.
Noting that in the revised manuscript, the edited and added contents are marked with green text. We did this to make the newly added parts easy to follow. We will of course return them back to the default color for the final version of the manuscript.
Round 2
Reviewer 1 Report
Thank you for addressing my comments thoroughly, especially the novelty issue. I am happy to accept this paper as a novel contribution in the area of large-scale PUF modeling.
Reviewer 3 Report
I am not an expert in electronic PUFs and I cannot judge the novelty of the work. Given that the authors have addressed all of the issues raised in my first report, I recommend publication of the manuscript.